# Younger generations are more interested than older generations in having non-domesticated animals as pets

Katherine A. Cronin[1]*, Maureen Leahy[1], Stephen R. Ross[2], Mandi Wilder Schook[3], Gina M. Ferrie[3], Andrew C. Alba[3]

1 Animal Welfare Science Program, Lincoln Park Zoo, Chicago, Illinois, United States of America, 2 Lester E. Fisher Center for the Study and Conservation of Apes, Lincoln Park Zoo, Chicago, Illinois, United States of America, 3 Animals, Science and Environment, Disney's Animal Kingdom®, Lake Buena Vista, Florida, United States of America

* kcronin@lpzoo.org

**Data Availability Statement:** All relevant data are within the manuscript and its Supporting Information files.

## Abstract

The trade and private ownership of non-domesticated animals has detrimental effects on individual animals and their wild populations. Therefore, there is a need to understand the conditions that motivate and dissuade interest in non-domesticated pet ownership. Past research has demonstrated that the way in which non-domesticated animals are portrayed in images influences the public's perception that they are suitable as pets. We conducted an online survey of people residing in the United States to investigate how viewing images that could be realistically captured in the zoo and broader tourism industries impact the degree to which people report interest in having that animal as a pet. We focused on two species, reticulated pythons (*Malayopython reticulatus*) and two-toed sloths (*Choloepus hoffmanni*), and presented each species in six different visual contexts. After viewing an image, respondents reported interest in pet ownership on a four-point Likert scale. Each species was studied separately in a between-subjects design and results were analyzed using ordinal logistic regression models. Thirty-nine percent of respondents reported interest in sloth pet ownership, and 21% reported interest in python pet ownership. However, contrary to our hypotheses, we found that viewing these species in different visual contexts did not significantly affect survey respondents' reported interest in having either species as a pet. Generation was a significant predictor of interest in both sloth and python pet ownership, with younger generations reporting more interest in having these species as pets. Male respondents reported more interest in python pet ownership, whereas there were no significant differences between genders regarding interest in sloth ownership. We consider how modern media exposure to animals in unnatural contexts may relate to the generational effect and discuss priorities for future research to better understand the development of individual interests in non-domesticated pet ownership.

**Funding:** The authors received no specific funding for this work.

**Competing interests:** The authors have declared that no competing interests exist.

## Introduction

The trade and private ownership of non-domesticated animals has detrimental effects on individual animals and their wild populations. Unfortunately, non-domesticated (or "non-traditional," or "exotic") pet ownership is widespread and the global market for non-domesticated pets, both legally-traded and illegally-trafficked, is increasing and negatively impacting the status of wild populations of birds, reptiles, amphibians, and mammals [1–3]. Non-domesticated pets are can be sourced from wild populations, known to threaten biodiversity and conservation efforts [1,2,4–8].

The welfare of individual animals who are part of this trade and trafficking is often negatively impacted as well. Regardless of their origin, animal welfare is typically compromised while the animals are in private homes or residing in poorly regulated business ventures (e.g., animal cafes or touristic photo opportunities). Animals in these environments often undergo painful procedures such as defanging or declawing to minimize the chance of injury to people. In many cases, these environments introduce behavioral restriction and the environments do not meet the animals' needs [e.g., 3,9–15, but see 16].

Furthermore, private ownership and unregulated illegal trade of animals may pose risks of disease transmission that are harmful to both humans and nonhuman animals if conditions are poorly managed. Diseases can be transmitted between animals in crowded conditions during capture, transport and sale, and from animals to humans during close contact [17–23]. Viruses that spread from animal to human hosts have been responsible for massive outbreaks of disease in humans, most recently demonstrated in the global COVID-19 pandemic [24,25]. With known deleterious effects of non-domesticated animal trade and trafficking on conservation efforts, animal welfare, and human and non-human animal health, there is a pressing need to understand the conditions that motivate and dissuade interest in non-domesticated animals as pets.

People have become more, rather than less, interested in non-domesticated pets in recent decades [3,26], and those purchasing these animals are often unaware that they may be contributing to illegal or harmful trade [6]. A recent survey found that people were more likely to be dissuaded from pet ownership due to threats of zoonotic disease and legal ramifications than due to concerns about loss of biodiversity or compromised animal welfare [27]. The increase in non-domesticated pet interest may be due to the widespread availability of the internet and social media, both because of the increased market access for those trying to reach consumers and because of the exposure that people have to images and videos of animals [28–35]. Further, public interest in non-domesticated pet ownership is known to be responsive to media portrayals of animals on Facebook, YouTube, and in popular movies and television shows [36–43].

Experimental research has also demonstrated that the way in which non-domesticated animals are portrayed influences the public's perception that they are suitable as pets. For example, undergraduate students shown videos of chimpanzees playing an entertaining role in commercials were more likely to answer positively on questions gauging the suitability of chimpanzees as pets than were students who saw videos of chimpanzees in a national park or as part of a conservation commercial [44]. Likewise, people shown images of chimpanzees with a person standing nearby were over 30% more likely to report chimpanzees to be appealing as a pet compared to people viewing an image of the same chimpanzee without a person nearby [45]. Another investigation that focused on three different primate species (two monkeys and one prosimian) revealed similar results and demonstrated that when primates were shown in an unnatural setting (an office workplace), people were more likely to consider them suitable pets [46]. Respondents were less likely to judge the animals to be suitable pets if there

was no human present, or if the primate was shown in a naturalistic, forested setting. Similar results were also found in a recent study surveying interest in pet ownership among visitors to a combination zoo-amusement park. In this study, visitors were shown images of several different species superimposed in five varying contexts, and interest in pet ownership was lowest in the only condition that did not have a human present in the image [31]. Taken together, these studies indicate that the environment or context in which non-domesticated animals are presented, and the presence of humans nearby or in contact, can affect the perception that non-domesticated animals are suitable pets. Some organizations have responded to this research by dissuading the sharing of images with characteristics known to encourage the perception of non-domesticated animals as pets [47–49].

The question of how to responsibly portray non-domesticated animals in a way that does not inadvertently encourage pet ownership is especially relevant for zoos where millions of people are regularly brought into close proximity with a diversity of non-domesticated animal species. Zoos adopt many strategies to connect people with wildlife. Many zoos offer visitors the opportunity to view animals in naturalistic zoo exhibits [50,51]. Additionally, many zoos facilitate up-close experiences with animals, and these experiences may or may not involve physical contact between visitors and the animals [52]. If and how these different experiences foster compassion for animals and promote pro-conservation behavioral changes in visitors is a topic of current interest [53–55]. These various opportunities to be close to wildlife generate visuals, both in real time and in the form of images shared interpersonally or widely on the internet, which may inadvertently influence interest in pet ownership. Many zoo professionals provide responsible interpretation and conservation messaging during these live opportunities, sometimes even directly advising against non-domesticated pet ownership. However, this messaging does not reach zoo visitors passing by nor the public viewing images on the internet later without context.

Here, we focus research efforts on these portrayal effects in relation to reticulated pythons (*Malayopython reticulatus*) and two-toed sloths (*Choloepus hoffmanni*). These two species are popular in zoos and commonly used in up-close experiences with visitors [52], their demand in the pet trade may negatively impact wild populations [56,57], and individuals of both species are likely to experience poor welfare in the private pet trade given their biological adaptations and specific husbandry needs. While the study was not designed to test for differences between the two species, we were interested in studying two species that differ in their attributes in a way that may modify people's attitudes toward them, including their perceived cuteness and vulnerability, and their physical similarity to humans [58]. The goal of the present study was to establish the prevalence of interest in pet ownership for these two species, and to question whether visual images that are commonly witnessed or photographed in zoo or tourist settings impact people's interest in having that animal as a pet. To do so, we presented online survey respondents with one of several different images showing the animals in varying environments and contexts. Based on past research, we hypothesized that people would report more interest in pet ownership after viewing images of animals in non-naturalistic environments and images of animals with humans in the image. Furthermore, given previous research showing effects of gender and age on animal attitudes [31,59–62], we also considered how gender and age impacted pet interest.

## Materials and methods

### Stimuli

Twelve visual stimuli were created for this experiment (Fig 1). One identical animal image was used to create all stimuli for that species. Different backgrounds were applied to portray each

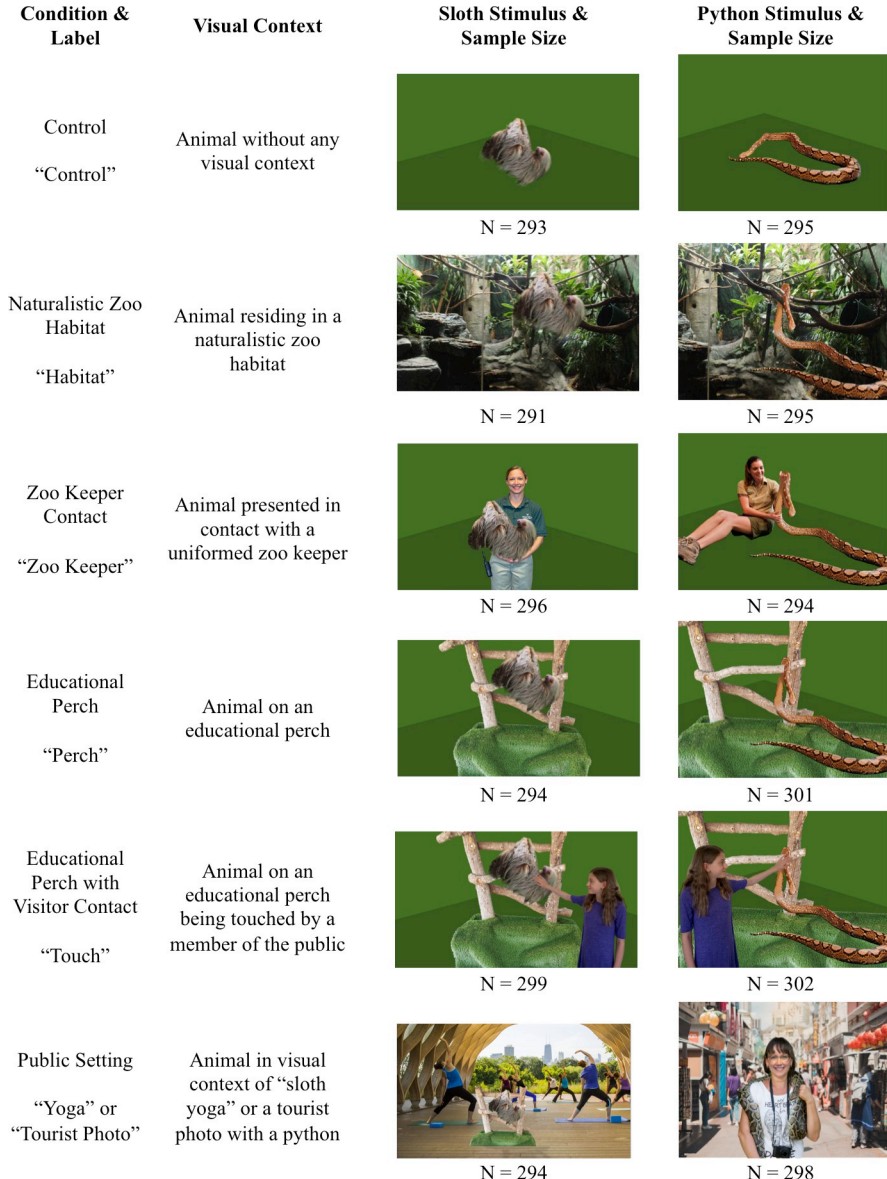

| Condition & Label | Visual Context | Sloth Stimulus & Sample Size | Python Stimulus & Sample Size |
|---|---|---|---|
| Control<br><br>"Control" | Animal without any visual context | N = 293 | N = 295 |
| Naturalistic Zoo Habitat<br><br>"Habitat" | Animal residing in a naturalistic zoo habitat | N = 291 | N = 295 |
| Zoo Keeper Contact<br><br>"Zoo Keeper" | Animal presented in contact with a uniformed zoo keeper | N = 296 | N = 294 |
| Educational Perch<br><br>"Perch" | Animal on an educational perch | N = 294 | N = 301 |
| Educational Perch with Visitor Contact<br><br>"Touch" | Animal on an educational perch being touched by a member of the public | N = 299 | N = 302 |
| Public Setting<br><br>"Yoga" or "Tourist Photo" | Animal in visual context of "sloth yoga" or a tourist photo with a python | N = 294 | N = 298 |

**Fig 1. Experimental conditions, visual stimuli, and sample size per condition.** Conditions presented in the online survey in a between-participants design, and number of subjects in each condition. An additional condition was run for internal evaluation of a Lincoln Park Zoo program; details are in Supporting Information.

species in the natural and unnatural contexts of interest, and a simple green background was applied to create control stimuli for each species. Stimuli materials were gathered from open-source photos or taken by the authors. The individuals pictured in Fig 1 have provided written informed consent (as outlined in PLOS consent form) to publish their images alongside the manuscript. Stimuli were created using Adobe Photoshop 2020 version 21.2.

## Survey design

Surveys were created using Qualtrics software, Version January 2021 (Qualtrics, Provo, UT, www.qualtrics.com) and administered online using Amazon's Mechanical Turk (MTurk)

(www.mturk.com), a website for conducting behavioral research that provides access to a large and diverse subject pool and pays participants [63]. Potential participants were recruited from MTurk's worker base. Workers resided in the United States and searched for our survey using keywords "survey," "attitude," "opinion," and/or "perception," read a brief description of the survey, and were notified of the compensation provided for completing the survey ($0.50) before agreeing to take the survey. Surveys were administered in a between-subjects design such that each respondent only saw one stimulus type.

The survey began with the presentation of one randomly selected stimulus (Fig 1) for 15 sec. Respondents could not advance to the survey statement until 15 sec had elapsed. There was no image presented concurrent with the survey statement. The statement they were shown was either "I would like to have a sloth as a pet," or "I would like to have a python as a pet," depending on the treatment to which they were randomly assigned. Responses were provided via a 4-point Likert scale with the terms "strongly disagree, disagree, agree, and strongly agree" as response choices. After the respondent made their selection, they were presented with an additional 23 questions that were the focus of a separate study and are not interpreted here. Finally, respondents entered demographic information including gender and age. There was no time limit for individual questions, and respondents had a maximum time of 20 minutes to complete the entire survey. Respondents were compensated if they completed the survey in full. Survey data collection occurred from Jan 11–13, 2021.

No animals were handled for the purposes of this study. This study was deemed exempt by the Lincoln Park Zoo Institutional Review Board (IRB-20-01-EX), approved by the Lincoln Park Zoo Research Committee, and approved by Disney's Animal Care and Welfare Committee (IR2004) and Scientific Review Committee.

## Data analysis

Survey responses were downloaded from Qualtrics, organized in Microsoft Excel, and analyzed in R version 3.6.3 [64]. Data were cleaned to exclude subjects who reported their age to be < 18 years (3 respondents), those who provided the same answer to all survey items (34 respondents), and those who completed the full survey in an unrealistic time frame (less than one minute; an additional 10 respondents). The resulting data set included 1785 respondents exposed to a python image and 1767 respondents exposed to a sloth image (see Fig 1 for details of sample size per condition).

Data were analyzed separately by species due to their different propensity to engender pet interest [58] and due to the impossibility of creating realistic stimuli that differed only by species. Survey responses were analyzed by ordinal logistic regression to estimate the effects of visual context, subject age and gender on interest in exotic pet ownership using the "polr" function in the R package MASS [65]. For both species, the dependent variable was a four-level ordered categorical variable (the respondent's Likert response), and the predictor variables were visual context (fixed categorical predictor with six levels, Fig 1), respondent age (fixed categorical predictor binned by generation following [66] as follows: Gen Z (age at time of survey 18–24), Millennial (age 25–40), Gen X (age 41–56), Boomers II (57–66), Boomers I (67–75), Post War (76–93)) and respondent gender (fixed categorical predictor: male, female, other). Although additional gender identification was provided by respondents (Supporting Information), these three categories were used for analyses following [67]. To determine whether predictors were significant, we used the "anova" function in the CAR package to perform Type II likelihood ratio tests on the ordinal logistic regression models. Assumptions were met for both the python and sloth models; there was no evidence of multi-collinearity and the Brant test [68] indicated that the parallel regression assumption held for both datasets. The

**Table 1. Proportion of responses by level of agreement with the statement, "I would like to have a sloth/python as a pet".**

|  | Strongly Agree | Agree | Disagree | Strongly Disagree |
|---|---|---|---|---|
| Sloth: All generations | 0.112 | 0.276 | 0.353 | 0.260 |
| Sloth: Gen Z | 0.054 | 0.137 | 0.327 | 0.482 |
| Sloth: Millennial | 0.046 | 0.198 | 0.261 | 0.495 |
| Sloth: Gen X | 0.053 | 0.157 | 0.225 | 0.564 |
| Sloth: Boomers II | 0.019 | 0.075 | 0.182 | 0.723 |
| Sloth: Boomers | 0.000 | 0.045 | 0.104 | 0.851 |
| Sloth: Post War | 0.000 | 0.000 | 0.000 | 1.000 |
| Python: All generations | 0.044 | 0.165 | 0.245 | 0.546 |
| Python: Gen Z | 0.134 | 0.282 | 0.401 | 0.183 |
| Python: Millennial | 0.133 | 0.305 | 0.332 | 0.229 |
| Python: Gen X | 0.089 | 0.256 | 0.379 | 0.275 |
| Python: Boomers II | 0.038 | 0.197 | 0.357 | 0.408 |
| Python: Boomers | 0.043 | 0.085 | 0.383 | 0.489 |
| Python: Post War | 0.000 | 0.167 | 0.333 | 0.500 |

Proportions of response selections per species and generation, collapsing across genders.

Brant tests were run using the function "brant" in the R package BRANT. Data were visualized using the R package effects [69,70].

## Results and discussion

Overall, 38.71% of respondents indicated agreement that they would like to have a sloth as a pet, and 20.96% of respondents indicated agreement that they would like to have a python as a pet (considering "agree" and "strongly agree" combined, Table 1). The results of likelihood ratio tests to identify whether model predictors were significant are reported in Table 2, and full model results are reported in Tables 3 and 4. Contrary to our hypothesis, the visual context in which the animal was presented was not a significant predictor of interest in pet ownership for sloths nor for pythons (Table 2, Figs 2A and 3A). Considering sloths, generation was a significant predictor of interest in pet ownership but gender was not (Tables 2 and 3, Fig 2B and 2C). Younger generations tended to agree more with the statement that they would like to have a sloth as a pet. Considering pythons, generation and gender were both significant

**Table 2. Likelihood ratio test results for ordinal logistic regression models.**

| Sloth Model | $\chi^2$ | df | P-value |
|---|---|---|---|
| Fixed Factors |  |  |  |
| Context | 3.872 | 5 | 0.5680 |
| Gender | 4.459 | 2 | 0.1076 |
| **Generation** | **59.676** | **5** | **1.418 e-11** |
| **Python Model** | **$\chi^2$** | **df** | **P-value** |
| Fixed Factors |  |  |  |
| Context | 5.897 | 5 | 0.3163 |
| **Gender** | **24.441** | **2** | **4.929 e-06** |
| **Generation** | **63.938** | **5** | **1.861 e-12** |

Results of the likelihood ratio tests isolating the contributions of the fixed effects in both the sloth and python models are shown in the table. The factors in bold are significant predictors of interest in pet ownership.

**Table 3. Ordinal logistic regression results for the sloth experiment.**

| Predictor | Coefficient | Lower-95 | Upper-95 | S.E. | Odds Ratio |
|---|---|---|---|---|---|
| Context_Naturalistic Zoo Habitat | -0.0323 | -0.3280 | 0.2633 | 0.1509 | 0.9682 |
| Context_Keeper Contact | -0.1580 | -0.4526 | 0.1363 | 0.1502 | 0.8538 |
| Context_Educational Perch | 0.0639 | -0.2317 | 0.3595 | 0.1508 | 1.0660 |
| Context_Educational Perch with Visitor Contact | 0.1100 | -0.1854 | 0.4055 | 0.1507 | 1.1163 |
| Context_Public Setting (Yoga) | -0.0346 | -0.3314 | 0.2621 | 0.1514 | 0.9660 |
| Gender_Male | 0.0119 | -0.1595 | 0.1833 | 0.0875 | 1.0120 |
| Gender_Other | -1.0316 | -2.035 | -0.0639 | 0.4962 | 0.3565 |
| Generation_ Millennial | -0.0298 | -0.3454 | 0.2857 | 0.1609 | 0.9707 |
| Generation_ Gen X | -0.3772 | -0.7194 | -0.0355 | 0.1744 | 0.6858 |
| Generation_Boomers II | -0.9426 | -1.3571 | -0.5305 | 0.2108 | 0.3896 |
| Generation_Boomers I | -1.3510 | -1.9704 | -0.7457 | 0.3116 | 0.2590 |
| Generation_Post War | -1.3556 | -3.0201 | 0.1433 | 0.7812 | 0.2578 |

The reference value for visual context was the control condition, the reference value for gender was female, and the reference value for generation was Gen Z. Original coefficients are scaled in terms of logs and we provide the exponentiated odds ratios as well.

predictors of interest in pet ownership (Tables 2 and 4, Fig 3B and 3C). Younger generations tended to agree more with the statement that they would like to have a python as a pet, and females reported less interest in python pet ownership than other genders (20.2% of females reported agreement or strong agreement compared to 32.5% of males and 66.7% of respondents of other genders).

This study reveals interest in non-domesticated pet ownership in the United States general population. Pythons were indicated to be pets of interest for 1 out of 5 survey respondents, and sloths for 2 out of 5 survey respondents. This is comparable to past studies that have surveyed pet interest in response to visual stimuli in the United States and the United Kingdom. In the only previous survey of the general public in the United States, Ross et al. [45] reported between 27–37% of respondents reporting interest in chimpanzee pet ownership across their treatment conditions. Leighty et al. [46] surveyed visitors on grounds at a zoo in the United

**Table 4. Ordinal logistic regression results for the python experiment.**

| Predictor | Coefficient | Lower-95 | Upper-95 | S.E. | Odds Ratio |
|---|---|---|---|---|---|
| Context_Naturalistic Zoo Habitat | -0.1501 | -0.4694 | 0.1691 | 0.1629 | 0.8606 |
| Context_Keeper Contact | -0.0696 | -0.3870 | 0.2477 | 0.1619 | 0.9327 |
| Context_Educational Perch | 0.1709 | -0.1359 | 0.4776 | 0.1565 | 1.1863 |
| Context_Educational Perch with Visitor Contact | 0.0059 | -0.3061 | 0.3179 | 0.1592 | 1.0060 |
| Context_Public Setting (Tourist Photo) | 0.1555 | -0.1548 | 0.4658 | 0.1583 | 1.1682 |
| Gender_Male | 0.4722 | 0.2882 | 0.6562 | 0.0939 | 1.6035 |
| Gender_Other | 1.0356 | 0.2167 | 1.8545 | 0.4178 | 2.8168 |
| Generation_ Millennial | 0.0168 | -0.2882 | 0.3219 | 0.1556 | 1.0170 |
| Generation_ Gen X | -0.1529 | -0.4916 | 0.1858 | 0.1728 | 0.8582 |
| Generation_Boomers II | -0.9140 | -1.3586 | -0.4693 | 0.2269 | 0.4009 |
| Generation_Boomers I | -1.6722 | -2.4033 | -0.9411 | 0.3730 | 0.1878 |
| Generation_Post War | -13.1193 | -495.2670 | 469.0285 | 245.9937 | 0.000 |

The reference value for visual context was the control condition, the reference value for gender was female, and the reference value for generation was Gen Z. Original coefficients are scaled in terms of logs and we provide the exponentiated odds ratios as well.

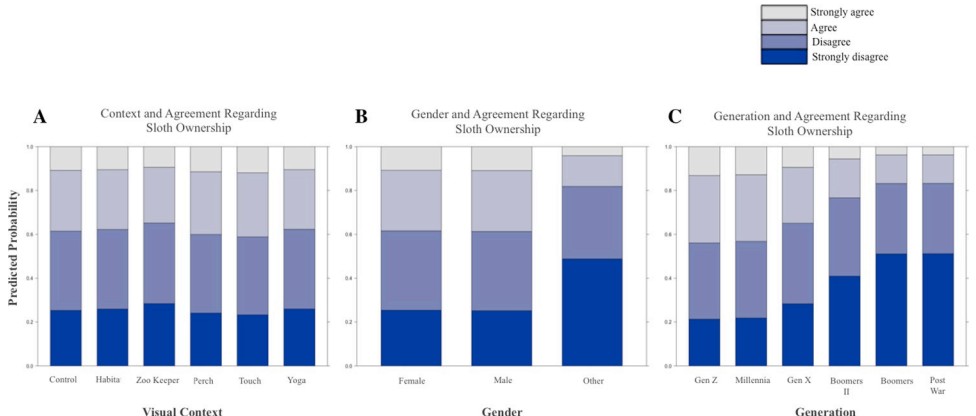

**Fig 2.** A-C. Predictor Effect Plots Showing the Role of Each Predictor on Interest in Sloth Ownership. Predictor effect plots provide graphical summaries for fitted regression models by averaging and conditioning the other predictor variables to summarize the role of a selected focal predictor in a fitted regression model (Fox & Weisberg, 2019). (Note: Figures sized to span two columns).

States and reported 16–21% of guests were interested in ownership of primate species. Finally, in a survey of visitors to a combination zoo-amusement park in the United Kingdom, Spooner & Stride [31] report 34–41% of respondents were interested in ownership of non-domesticated species. Understanding how these patterns of ownership interest differ across geographic regions, which vary in both regulations regarding non-domestic pet ownership and cultural attitudes about animals [71,72], remains to be determined.

We attempted to understand how the visual context in which an animal is portrayed influences interest in pet ownership. We showed images of sloths and pythons in naturalistic zoo enclosures, with professional animal care staff, with members of the public, outside of zoo habitats on educational perches, and in typical photo-prop tourism settings. Contrary to our hypotheses, and to previous studies of nonhuman primate species, we found no effect of visual context on interest in pet ownership. For both the sloth and python investigations, there was insufficient evidence that context affected the choice that viewers made regarding their interest in ownership of this species.

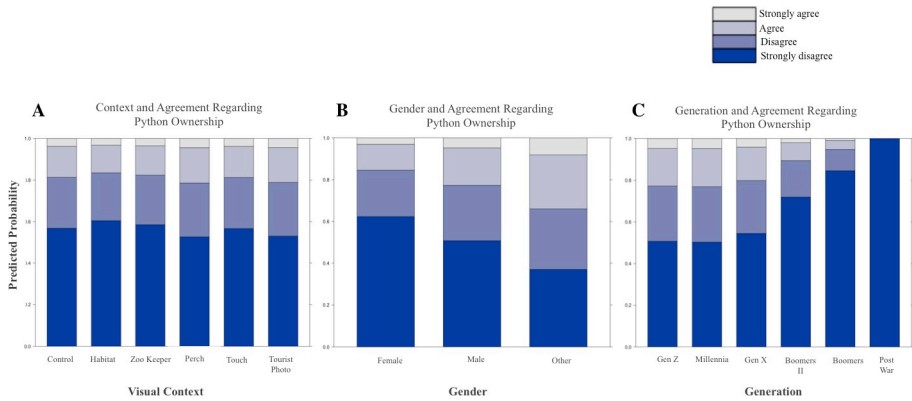

**Fig 3.** A-C. Predictor Effect Plots Showing the Role of Each Predictor on Interest in Python Ownership. Predictor effect plots provide graphical summaries for fitted regression models by averaging and conditioning the other predictor variables to summarize the role of a selected focal predictor in a fitted regression model (Fox & Weisberg, 2019). (Note: Figures sized to span two columns).

A key finding in the present study is that survey respondents from younger generations reported more agreement that they would like to have both a sloth and a python as a pet. This result was obtained for both species despite the species being considered independently in the study design, and despite the differences in their biology and appearance. Given the previous connections identified between media and interest in pet ownership [3,36–43], one explanation for these findings is that younger generations, which are regularly exposed to images of these animals in unnatural contexts and alongside humans in social media, are developing more interest in pet ownership than older generations without similar long-term exposure.

Given the dampened interest among older generations observed here, it is also possible that interest in non-domestic pet ownership wanes with age. Previous work has found that younger individuals show more interest and affection for animals than older individuals [73]. However, with the current cross-sectional (rather than longitudinal) study design, we cannot determine how interest in pet ownership may change with age and whether there are developmental effects influencing the patterns observed here. Certainly, portrayals of animals in unnatural contexts and alongside humans, for example in movie posters and in circuses, were not absent from the lives of the older generations studied. It is possible that the same older subjects sampled here would have expressed greater interest earlier in their lives, and similarly that the younger generations reporting greater interest may show declining interest with age. However, given the relationships reported between media and pet interest [36–43], we find it most plausible that interest reported among younger generations persists to some degree through development unless there are concurrent changes in media exposure.

We adopted a statement used in previous research used to understand interest in pet ownership, specifically, "I would like to have a sloth (or python) as a pet." Of course, agreement with statements about animals making appropriate pets, or statements about being interested in pet ownership, does not equate with actual pet ownership. Presumably, the vast majority of people surveyed who report interest in owning animals in the present study, or in previous studies [31,44–46] do not currently own these pets, and therefore there is some disconnect between the interest stated in response to the question and actions taken to pursue pet ownership. A future study that considers why people stating interest in ownership do not actually own the animals could be useful for identifying strategies to further curtail non-domestic pet ownership. It would also be useful to determine whether there is a predictable relationship between the frequency of stated measures of interest and the frequency of ownership. Regardless, given the negative impact of non-domesticated pet ownership on animal welfare and conservation discussed above, these numbers highlight the need to continue to study how to influence the perception that non-domesticated species are interesting pets.

We recognize that the lack of relationship between visual context and interest in non-domestic pet ownership may be due to the circumstances of this experiment. In reality, exposure to animals in varying contexts is a recurrent event in one's life, and degree of exposure depends largely on personal experiences and social and other media practices. The brief, 15-second exposure to the image of interest may not have been substantial enough to influence perception about the animal in light of respondents' own histories. Additionally, although past studies have detected differences in interest in pet ownership with brief exposure to images [45,46], it is possible that this specific online survey format in which the stimuli preceded the question did not measurably influence respondents' perceptions in the same way that other study designs have (e.g., those that used physical, in-hand photographs, or those that were available for persistent viewing throughout the survey).

Previous researchers have errantly interpreted a lack of significant effect of context on interest in non-domesticated pet ownership as indication that there is no influence of context [31]. When analyses fail to demonstrate statistical significance, with typical inferential statistics such

as used here and in Spooner & Stride [31], it is not statistically nor logically warranted to conclude that two treatments produce the same effect [e.g., 74]. All one can conclude is that the sample did not provide sufficient evidence to conclude that the effect exists. Therefore, unlike Spooner & Stride [31], we do not recommend that our findings be interpreted as evidence that humans in photos do not influence viewers' interest in sloth and python pet ownership. Rather, we acknowledge that additional research and alternative study designs are needed to understand precisely what types of visual experiences engender and discourage interest in non-domesticated pet ownership.

The present study design was intended to measure interest in non-domesticated pet ownership that is associated with viewing other's experiences with animals in educational programs or tourist opportunities. The stimuli in this study are reminiscent of images that would be seen in a social media feed or witnessed as one viewed another person participating in an animal experience. The stimuli do not replicate experiences that people themselves have in an in-person program, such as those experienced at zoos and aquaria. As such, respondents were not exposed to responsible interpretation or conservation-oriented messaging that characterizes animal programs at many accredited zoos. If and how verbal messaging interacts with visual context effects in zoo-based programs is a topic of current study [recently reviewed in 55,75,76], but not one tackled here.

With regards to gender, most previous research has demonstrated increased affinity and more positive attitudes toward animals by females compared to males [60,77–80]. In general, females tend to place more value on and show more concern for nonhuman species and conservation compared to males, and therefore report more positive wildlife attitudes [77,81]. Females may also form stronger emotional connections to individual animals than males [60]. However, for species associated with common phobias and predators, females have reported stronger fears, feelings of disgust, and negative attitudes associated with those particular animals [59,60,82,83]. An affinity for animals and propensity for positive wildlife attitudes may not equate to a desire to have a non-domesticated pet, however, as here we find no gender difference for interest in sloths as pets. We did find the least amount of interest in python ownership among females, which is consistent with stronger phobias reported by females towards predatory species.

The negative influence that non-domesticated pet ownership has on animal welfare and animal conservation is well established. What remains to be determined is how interest in non-domesticated pet ownership develops, and how it can be modulated through decisions that individuals and animal organizations make about both the in-person experiences that they create and the visuals that are associated with these experiences. Designing experiences for people that promote responsible actions and attitudes toward animals and avoid behavioral spillover of generating animal interest that can lead to unintended negative consequences for animals is important as organizations continue to develop ways to connect people to nature [55]. Given the prevalence of interest in non-domesticated pet ownership among the younger generations measured here, there is a pressing need for additional research with powerful alternative methodologies to better understand the human psychology driving this interest.

## Conclusion

The trade and private ownership of non-domesticated animals has detrimental effects on the welfare of individual animals and the conservation of their wild populations, and this study sought to identify whether brief exposure to common visual portrayals of sloths or pythons contributed to an interest in ownership of these species as pets. Thirty-nine percent of respondents reported interest in sloth pet ownership, 21% reported interest in python pet ownership,

and visual context did not significantly affect survey respondents' reported interest in having either species as a pet. However, generation was a significant predictor of interest in both sloth and python pet ownership, with younger generations reporting more interest in having both of these species as pets. We speculate that the regular media exposure to animals in unnatural contexts that is regularly experienced by younger generations may contribute to the generational effect found here and encourage further research to better understand the ontogeny of interest in non-domesticated pet ownership.

## Supporting information

**S1 File. This file contains several sections, including a table of participants' self-reported genders (S1 Table), information regarding an additional context evaluated for sloths (S1 Text and S1 Fig), and two tables (S2 Table and S3 Table) and a figure (S2 Fig) conveying the results of the analyses for the additional context.**
(DOCX)

**S1 Dataset. SlothData_Deposit.** This csv file contains the sloth survey data analyzed in the main text.
(CSV)

**S2 Dataset. PythonData_Deposit.** This csv file contains the python survey data analyzed in the main text.
(CSV)

**S3 Dataset. SlothDataLWL_Deposit.** This csv file contains the additional sloth data for the "Lettuce With Luigi" condition analyzed in the Supporting Information.
(CSV)

## Acknowledgments

We are grateful to Chris Bijalba for creating images, Jen Torchalski for assistance with image design, Lily Maynard and John Andrews for advice on survey design, and S. Sunny Nelson for feedback on the manuscript. We also thank three anonymous reviewers for their helpful feedback. We acknowledge the open-source sloth photo created by senivpetro and obtained from www.freepik.com.

## Author Contributions

**Conceptualization:** Katherine A. Cronin, Maureen Leahy, Stephen R. Ross, Mandi Wilder Schook, Gina M. Ferrie, Andrew C. Alba.

**Data curation:** Andrew C. Alba.

**Formal analysis:** Katherine A. Cronin, Andrew C. Alba.

**Methodology:** Katherine A. Cronin, Maureen Leahy, Stephen R. Ross, Mandi Wilder Schook, Gina M. Ferrie, Andrew C. Alba.

**Visualization:** Katherine A. Cronin.

**Writing – original draft:** Katherine A. Cronin, Maureen Leahy, Stephen R. Ross, Mandi Wilder Schook, Gina M. Ferrie, Andrew C. Alba.

**Writing – review & editing:** Katherine A. Cronin, Maureen Leahy, Stephen R. Ross, Mandi Wilder Schook, Gina M. Ferrie, Andrew C. Alba.

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
