## [Decision Letter · Decision Letter 0]

12 Oct 2021

PONE-D-21-28115Younger generations show persistent interest in non-domesticated animals as petsPLOS ONE

Dear Dr. Cronin,

Thank you for submitting your manuscript to PLOS ONE. After careful consideration, we feel that it has merit but does not fully meet PLOS ONE’s publication criteria as it currently stands. Therefore, we invite you to submit a revised version of the manuscript that addresses the points raised during the review process.

Many thanks for submitting your manuscript to PLOS One

It was reviewed by two experts in the field, and they have recommended some modifications be made prior to acceptance

I therefore invite you to make these changes and to write a response to reviewers which will expedite revision upon resubmission

I wish you the best of luck with your modifications

Hope you are keeping safe and well in these difficult times

Thanks

Simon

We look forward to receiving your revised manuscript.

Kind regards,

Simon Clegg, PhD

Academic Editor

PLOS ONE

2. We note that Figure 1 includes images of participants in the study. As per the PLOS ONE policy (http://journals.plos.org/plosone/s/submission-guidelines#loc-human-subjects-research) on papers that include identifying, or potentially identifying, information, the individual(s) or parent(s)/guardian(s) must be informed of the terms of the PLOS open-access (CC-BY) license and provide specific permission for publication of these details under the terms of this license. Please download the Consent Form for Publication in a PLOS Journal (http://journals.plos.org/plosone/s/file?id=8ce6/plos-consent-form-english.pdf). The signed consent form should not be submitted with the manuscript, but should be securely filed in the individual's case notes. Please amend the methods section and ethics statement of the manuscript to explicitly state that the patient/participant has provided consent for publication: “The individual in this manuscript has given written informed consent (as outlined in PLOS consent form) to publish these case details”. 

3. We note that Figure S1 in your submission contain copyrighted images. All PLOS content is published under the Creative Commons Attribution License (CC BY 4.0), which means that the manuscript, images, and Supporting Information files will be freely available online, and any third party is permitted to access, download, copy, distribute, and use these materials in any way, even commercially, with proper attribution. For more information, see our copyright guidelines: http://journals.plos.org/plosone/s/licenses-and-copyright.

a) You may seek permission from the original copyright holder of Figure S1 to publish the content specifically under the CC BY 4.0 license. 

Reviewers' comments:

Reviewer's Responses to Questions

**Comments to the Author**

1. Is the manuscript technically sound, and do the data support the conclusions?

Reviewer #1: Yes

Reviewer #2: Yes

Reviewer #3: Yes

2. Has the statistical analysis been performed appropriately and rigorously? 

Reviewer #1: Yes

Reviewer #2: Yes

Reviewer #3: Yes

3. Have the authors made all data underlying the findings in their manuscript fully available?

Reviewer #1: Yes

Reviewer #2: Yes

Reviewer #3: Yes

4. Is the manuscript presented in an intelligible fashion and written in standard English?

Reviewer #1: Yes

Reviewer #2: Yes

Reviewer #3: Yes

5. Review Comments to the Author

Reviewer #1: I congratulate the authors on an exceptionally well-written paper, with my recommendation that it should be accepted as is - with a few minor amendments to grammar and some suggestions for extra discussion points.

I have uploaded a PDF of the article with my comments attached for grammatical changes. The important suggestions are listed below line-by-line:

L58: Is there scope here to mention that often mutilation (de-fanging, de-clawing) is required as well?

L79: I think it would be interesting to add a sentence or two here about the apparent decrease in non-domesticated animal use in traditional film and television (e.g. reduced ape use) but the stark increase in YouTube, TikTok, and other social media videos with non-domesticated animals (usually illegal, like slow loris videos etc.) I found this recent article that explores some of the traditional media animal usages around the globe, might be relevant? Hitchens, P. L., Booth, R. H., Stevens, K., Murphy, A., Jones, B., & Hemsworth, L. M. (2021). The Welfare of Animals in Australian Filmed Media. Animals, 11(7), 1986. MDPI AG. Retrieved from http://dx.doi.org/10.3390/ani11071986

L236: Would be great to mention that there are possibly differences between US values and attitudes to those of public in other countries where exotic pet ownership is mostly illegal. i.e. USA has significant differences in terms of legal and illegal types of pets, whereas Australia and NZ have much stricter rules around legal pets, with most exotics banned (even some domesticated exotics) and the only way to get them is illegally. There is a possible cultural difference in terms of desire for or even exposure to these animals as pets too - you could mention that this is an important possible cross-cultural effect that needs to be studied more?

Reviewer #2: I enjoyed reading this manuscript. I think the submit matter is really interesting it’s written in a clear and understandable way. In the manuscript and analyses within, the author investigate whether the context in which an animal is viewed in an online picture affects the propensity of the viewer to want that animal as a pet. The hypothesis, based on previous studies (largely primate-based), was that photos with humans in them, or with humans handling the animals, would be more likely to elicit a wish for the animal as a pet compared to pictures in which the animal was in its natural surroundings or in an abstract form. There was insufficient evidence from this experiment to accept this hypothesis: the context of the animal in the picture made no significant difference to the answer obtained in response to the question “I would like to have a sloth/python as a pet”. However, males were more likely to want pythons as a pet, and for both sloths and reticulated pythons (the two species included in this study), there was an effect of age: younger generations were more likely to say “yes” ( 3 or 4 on the Likert scale) and this tendency decreased consistently with age (generational category).

I have a few comments, questions and suggestions that I hope will be constructive in this review, some minor, some edits, some opinions. These follow below:

BURYING THE LEAD

I think you bury the lead a bit too much. In its current form the manuscript very much focuses on whether the context of the picture affects the likelihood of an answer along the lines of ‘yes I would like a ___’ as a pet. This is understandable as it was presumably the main thrust of the experiment. However, there are a couple of thoughts I have on altering this to highlight different aspects of the analyses.

First, I think the finding that generation has a consistent effect to be really interesting. Younger participants were more likely to response “Yes”. I think the manuscript at present ‘Buries the Lead’ a little bit on this. I’d like to see this part of the experiment emphasized a little more. A few ways the authors could do this:

- Alter table 1 so that each row is subdivided within species to include generational category. This would help the reader see the actual data underlying this trend. So the row names would be something like: “Sloth Millenial; Sloth Generation X; Sloth Boomer 2; etc” and within each there’d be the proportion that answered within each Likert category.

- There is a little too much emphasis in the results/discussion on the negative result (i.e. the fact that context didn’t affect the likelihood of wanting the animal as a pet). Of course, it’s important to discuss this, but at present there is a really long passage of text (lines 238-275) that is primarily aimed at explaining why the apparent lack of effect of context in this study may be due to the study design, rather than the possibility that context doesn’t have an effect on people’s choices. I think it’s perfectly suitable to mention these possibilities (online photos Vs physical photos; 15 second exposure Vs longer exposure). It’s also important to highlight that absence of evidence isn’t evidence of absence, i.e. interpret these results with care. However, there is insufficient evidence that context affects the choice in these two species, and that needs stating clearly. At present it feels like a lot of words are dedicated to saying “we found no evidence of context impacting choices, but here’s why”. I think with a little editing of that section and some more neutral phrasing the main points in there are good, though.

- In contrast to the amount of text dedicated to explaining the ‘negative result’ (~40 lines), the main hook (in my opinion), that age did matter was somewhat buried. It was covered in only 12 lines in the discussion (Line 277-289). I feel like this and its implications could have more space devoted to them.

- Similarly, another point that is relevant to the above is that this is consistent for two very different species. It looks like much of the previous research in this area has been conducted on primates, but it’s very interesting to me that there’s no evidence that context matters in these other species, but that even so, age/generation does matter. It’d be interesting to hear more on how general this could be and how, with the use of animals on social media, what the implications could be.

FRAMING OF THE QUESTION

I’d like to see a little more justification for the framing of the question and how it was presented to the participants. The authors mention in the discussion that “Of course, agreement with statements about animals making appropriate pets, or statements about being interested in pet ownership, does not equate one-to-one with actual pet ownership.” Is there any evidence from other papers or from this one of how this relationship works, or are there techniques to be able to account for the possibility of different interpretations? I think the question is problematic for me because although it seems really straightforward, it could be interpreted in different ways. For example: “I would like to have a python as a pet” seems pretty easy to answer on an operational basis, but it could also be interpreted on a more theoretical level. If some kind of negative control were included that were not operationally feasible in most instances, that could tell us something about the level of ‘realistic’ answers obtained.

For example: “I would like to have a rhinoceros as a pet” or “I would like to have a unicorn as a pet” might seem facetious as they’re operationally (or completely) impossible for most people but there might be a baseline level of people who would still see a picture of one and say “I want a rhino as a pet”.

I ask this and wonder if it’s possible to calibrate the results in this way because having spent time in zoos you do hear those kinds of comments, but I struggle to believe people really mean that they actually want a pet rhino/elephant/etc (although people constantly surprise me).

LEVEL OF POSITIVE RESPONSES

Regardless of exactly how the question is interpreted, I think it is important to point out more clearly that the absolute interest in these animals as pets was low regardless of context (~20% for reticulate pythons in particular). This is important for understanding and managing demand – if it is a relatively small amount of the population that means measures to restrict sales/education potential buyers can be more focused than if the proportions were higher.

MORE DETAIL ON WILDLIFE TRADE

I think at present the way the wildlife trade for non-domesticated animals is too coarse. It’s a really complex, subtle subject matter, and the way the manuscript outlines on numerous occasions that its negative effects are clear/well-documented could contain more relevant detail to help the reader understand what those negative impacts are. Below I’ve listed some examples that I think would benefit from more detail. The quotation from the paper is in speech marks, my response/suggestion next to it.

“A considerable proportion of non-domesticated pets are sourced from wild populations, threatening biodiversity and conservation efforts [1, 2, 4-8]”.

Increasingly in many countries, the most commonly kept animals that are legally kept are from captive bred sources. What proportion is a considerable proportion (varies by region probably?)? Can you give any examples that are relevant to the current analyses?

“Welfare, or the quality of life as experienced by the animal, is threatened when animals are captured from the wild, both during transport into captivity, and while they are in private homes or poorly regulated business ventures (e.g., animal cafes or touristic photo opportunities)”.

Key point for me: not only from wild-caught animals either – this is the case for any animal in the trade, including captive-bred ones. In fact, for many species in the wildlife trade in many countries the trade doesn’t necessarily cause conservation concern, but it still contains many welfare issues. For example, CITES listed species, although not perfectly monitored are generally sustainably managed, and species that aren’t CITES-listed in most cases are not listed because their population numbers aren’t threatened by trade. However, there are other issues here such as species becoming invasives and damaging native ecosystems, spread of pathogens, and in my mind most commonly and importantly, regardless of the conservation status of a species welfare issues at many points in the supply chain are a concern. I’d like to see more detail in the statements on “wildlife trade is bad” to provide evidence of how it’s bad, and in what contexts (ideally balanced with some examples of why it’s popular and also has benefits (the cost-benefit of any animal ownership, domesticated or not has many issues, not least welfare)).

[this reference has some good balanced arguments on the non-domestics trade: Pasmans, F., Bogaerts, S., Braeckman, J., Cunningham, A.A., Hellebuyck, T., Griffiths, R.A., Sparreboom, M., Schmidt, B.R. and Martel, A., 2017. Future of keeping pet reptiles and amphibians: towards integrating animal welfare, human health and environmental sustainability. Veterinary Record, 181(17), pp.450-450.]

“Furthermore, legal or illegal ownership of non-domesticated animals poses risks of disease transmission that are harmful both to humans and nonhuman animals”.

As above - this risk is almost certainly greater in the ownership and transport of domesticated animals. That doesn’t negate the above point, but needs acknowledging.

“their demand in the pet trade is negatively impacting wild populations [50, 51], and individuals of both species are likely to experience poor welfare in the private pet trade [51, 52].”

Reference 52 is about ball pythons, not reticulated pythons. These species are greatly different in their geographic range (West Africa Vs. S.E. Asia), and size (1.2-2m Vs 3m+). That being the case, the reference to poor welfare needs changing as these are different species with very different processes in the trade.

Further, there is actually empirical evidence that reticulated pythons can be managed for general trade in a sustainable fashion – see Natusch, D.J., Lyons, J.A., Riyanto, A. and Shine, R., 2016. Jungle giants: assessing sustainable harvesting in a difficult-to-survey species (Python reticulatus). PLoS One, 11(7), p.e0158397. for details

Again, that doesn’t mean there aren’t welfare issues with reticulated pythons in the trade or the hobby (they are very large animals that have specific husbandry needs), but the above points don’t sufficiently support the point being made.

Reviewer #3: After reading the paper, I have recommended that it be resubmitted pending minor alterations. While the conclusions drawn about the variation between males and females is justifiable, it is my opinion that the discussion pertaining to the difference between generations requires expansion.

As it is the primary focus of the paper, as stated in the paper title, the reasons for the results seen need to be presented in further detail. Considering the use of animals in media for decades, as well as other means of contact (circus' etc.), older generations would also have been exposed to substantial portrayals of non-domesticated animals in unnatural settings. It is for this reason that I recommend that additional explanations be presented to strengthen the point.

Minor issues:

Figures 2 and 3 are blurred and subsequently, difficult to read.

When referring to Figures 1, 2 and 3 in the text, they ought to be referred to specifically (e.g. Fig. 3A) when discussing specific elements of the figure.

Overall, I thought that the paper touched on a really interesting and important issue, and provides a strong foundation for the continuation of the investigation. I particularly appreciated the species chosen, to reflect the potential difference between animals with and without anthropomorphic characteristics.

6. PLOS authors have the option to publish the peer review history of their article (what does this mean?). If published, this will include your full peer review and any attached files.

Reviewer #1: No

Reviewer #2: No

Reviewer #3: No

---

## [Author Response · Author response to Decision Letter 0]

24 Nov 2021

(The text below is the same as can be found in the "Response to Reviewers" file uploaded with this resubmission.)

Response to Reviewers 

All line numbers refer to numbering in the “Manuscript” file with tracked changes accepted.

Comments regarding the journal requirements

COMMENT: Please ensure that your manuscript meets PLOS ONE's style requirements, including those for file naming. The PLOS ONE style templates can be found at 

OUR RESPONSE: We ensure our manuscript meets these requirements.

COMMENT: We note that Figure 1 includes images of participants in the study. As per the PLOS ONE policy (http://journals.plos.org/plosone/s/submission-guidelines#loc-human-subjects-research) on papers that include identifying, or potentially identifying, information, the individual(s) or parent(s)/guardian(s) must be informed of the terms of the PLOS open-access (CC-BY) license and provide specific permission for publication of these details under the terms of this license. Please download the Consent Form for Publication in a PLOS Journal (http://journals.plos.org/plosone/s/file?id=8ce6/plos-consent-form-english.pdf). The signed consent form should not be submitted with the manuscript, but should be securely filed in the individual's case notes. Please amend the methods section and ethics statement of the manuscript to explicitly state that the patient/participant has provided consent for publication: “The individual in this manuscript has given written informed consent (as outlined in PLOS consent form) to publish these case details”. If you are unable to obtain consent from the subject of the photograph, you will need to remove the figure and any other textual identifying information or case descriptions for this individual.

OUR RESPONSE: We have obtained signed consent forms which we have on file, and have added the statement to the methods. 

COMMENT: We note that Figure S1 in your submission contain copyrighted images. All PLOS content is published under the Creative Commons Attribution License (CC BY 4.0), which means that the manuscript, images, and Supporting Information files will be freely available online, and any third party is permitted to access, download, copy, distribute, and use these materials in any way, even commercially, with proper attribution. For more information, see our copyright guidelines: http://journals.plos.org/plosone/s/licenses-and-copyright. We require you to either (1) present written permission from the copyright holder to publish these figures specifically under the CC BY 4.0 license, or (2) remove the figures from your submission: 

a) You may seek permission from the original copyright holder of Figure S1 to publish the content specifically under the CC BY 4.0 license. 

We recommend that you contact the original copyright holder with the Content Permission Form (http://journals.plos.org/plosone/s/file?id=7c09/content-permission-form.pdf) and the following text: “I request permission for the open-access journal PLOS ONE to publish XXX under the Creative Commons Attribution License (CCAL) CC BY 4.0 (http://creativecommons.org/licenses/by/4.0/). Please be aware that this license allows unrestricted use and distribution, even commercially, by third parties. Please reply and provide explicit written permission to publish XXX under a CC BY license and complete the attached form.”

OUR RESPONSE: We have received permission from the copyright holder and uploaded this permission form. We have added the text as advised to the figure caption.

Reviewers' comments

Reviewer #1

COMMENT: I congratulate the authors on an exceptionally well-written paper, with my recommendation that it should be accepted as is - with a few minor amendments to grammar and some suggestions for extra discussion points. I have uploaded a PDF of the article with my comments attached for grammatical changes. The important suggestions are listed below line-by-line:

OUR RESPONSE: Thank you. We have revised the manuscript text in accordance with all of the comments in the pdf.

COMMENT: L58: Is there scope here to mention that often mutilation (de-fanging, de-clawing) is required as well?

OUR RESPONSE: Yes, we have added mention of these procedures.

COMMENT: L79: I think it would be interesting to add a sentence or two here about the apparent decrease in non-domesticated animal use in traditional film and television (e.g. reduced ape use) but the stark increase in YouTube, TikTok, and other social media videos with non-domesticated animals (usually illegal, like slow loris videos etc.) I found this recent article that explores some of the traditional media animal usages around the globe, might be relevant? Hitchens, P. L., Booth, R. H., Stevens, K., Murphy, A., Jones, B., & Hemsworth, L. M. (2021). The Welfare of Animals in Australian Filmed Media. Animals, 11(7), 1986. MDPI AG. Retrieved from http://dx.doi.org/10.3390/ani11071986

OUR RESPONSE: Thank you for this idea and for calling our attention to Hitchens et al. After reading it, it does not seem to provide information about decreases in non-domestic animal use in film/television in contrast to social media as suggested in this comment, and we are unable to find a source that can document the pattern the reviewer is describing. However, the commentary does consider thoughtfully the same references that we include in the current manuscript, coming to similar conclusions about the threat of portrayals on pet interest. We have now referenced it on line 79.

COMMENT: L236: Would be great to mention that there are possibly differences between US values and attitudes to those of public in other countries where exotic pet ownership is mostly illegal. i.e. USA has significant differences in terms of legal and illegal types of pets, whereas Australia and NZ have much stricter rules around legal pets, with most exotics banned (even some domesticated exotics) and the only way to get them is illegally. There is a possible cultural difference in terms of desire for or even exposure to these animals as pets too - you could mention that this is an important possible cross-cultural effect that needs to be studied more?

OUR RESPONSE: Yes, agree! We have clarified in the para beginning on line 225 that the results discussed here came primarily from studies in the US. We have now added information about the value of expanding this work to consider how legislation (directly or indirectly) and attitudes about animals (at an individual or cultural level) influence in interest in non-domestic pet ownership.

Reviewer #2

COMMENT: I enjoyed reading this manuscript. I think the submit matter is really interesting it’s written in a clear and understandable way. In the manuscript and analyses within, the author investigate whether the context in which an animal is viewed in an online picture affects the propensity of the viewer to want that animal as a pet. The hypothesis, based on previous studies (largely primate-based), was that photos with humans in them, or with humans handling the animals, would be more likely to elicit a wish for the animal as a pet compared to pictures in which the animal was in its natural surroundings or in an abstract form. There was insufficient evidence from this experiment to accept this hypothesis: the context of the animal in the picture made no significant difference to the answer obtained in response to the question “I would like to have a sloth/python as a pet”. However, males were more likely to want pythons as a pet, and for both sloths and reticulated pythons (the two species included in this study), there was an effect of age: younger generations were more likely to say “yes” ( 3 or 4 on the Likert scale) and this tendency decreased consistently with age (generational category).

I have a few comments, questions and suggestions that I hope will be constructive in this review, some minor, some edits, some opinions. These follow below:

BURYING THE LEAD

I think you bury the lead a bit too much. In its current form the manuscript very much focuses on whether the context of the picture affects the likelihood of an answer along the lines of ‘yes I would like a ___’ as a pet. This is understandable as it was presumably the main thrust of the experiment. However, there are a couple of thoughts I have on altering this to highlight different aspects of the analyses.

OUR RESPONSE: Thank you for your positive assessment. Indeed the emphasis on the interest in pet ownership as it related to visual treatment was the primary motivation for the experiment and probably explains why we over-emphasize those results in relation to the generational effect you highlight. We have modified the manuscript to emphasize the generational effects in several ways, described in our responses below.

COMMENT: First, I think the finding that generation has a consistent effect to be really interesting. Younger participants were more likely to response “Yes”. I think the manuscript at present ‘Buries the Lead’ a little bit on this. I’d like to see this part of the experiment emphasized a little more. A few ways the authors could do this:

Alter table 1 so that each row is subdivided within species to include generational category. This would help the reader see the actual data underlying this trend. So the row names would be something like: “Sloth Millenial; Sloth Generation X; Sloth Boomer 2; etc” and within each there’d be the proportion that answered within each Likert category.

OUR RESPONSE: Thank you for this suggestion, we’ve updated Table 1 as suggested.

COMMENT: There is a little too much emphasis in the results/discussion on the negative result (i.e. the fact that context didn’t affect the likelihood of wanting the animal as a pet). Of course, it’s important to discuss this, but at present there is a really long passage of text (lines 238-275) that is primarily aimed at explaining why the apparent lack of effect of context in this study may be due to the study design, rather than the possibility that context doesn’t have an effect on people’s choices. I think it’s perfectly suitable to mention these possibilities (online photos Vs physical photos; 15 second exposure Vs longer exposure). It’s also important to highlight that absence of evidence isn’t evidence of absence, i.e. interpret these results with care. However, there is insufficient evidence that context affects the choice in these two species, and that needs stating clearly. At present it feels like a lot of words are dedicated to saying “we found no evidence of context impacting choices, but here’s why”. I think with a little editing of that section and some more neutral phrasing the main points in there are good, though.

OUR RESPONSE: We agree and thank you for calling our attention to this. We have now added another statement to clearly state that there was insufficient evidence for both species (line 244) and moved that statement earlier in the discussion. We have separated that statement from the discussion of why we may not have found an effect (now starting on line 286), which now comes after the discussion of the generational effect. 

COMMENT: In contrast to the amount of text dedicated to explaining the ‘negative result’ (~40 lines), the main hook (in my opinion), that age did matter was somewhat buried. It was covered in only 12 lines in the discussion (Line 277-289). I feel like this and its implications could have more space devoted to them.

OUR RESPONSE: Agreed. We have de-emphasized the explanation of the negative result as explained in response to the previous comment and expanded upon the implications of the generational effect, see lines 248-269. We have also modified the title and moved the reporting of the generational effect earlier in the abstract and discussion to reflect this increased emphasis.

COMMENT: Similarly, another point that is relevant to the above is that this is consistent for two very different species. It looks like much of the previous research in this area has been conducted on primates, but it’s very interesting to me that there’s no evidence that context matters in these other species, but that even so, age/generation does matter. It’d be interesting to hear more on how general this could be and how, with the use of animals on social media, what the implications could be.

OUR RESPONSE: We agree, and we have now added more discussion of the consistent results obtained for the two independent species analyses (line 250) and more discussion of the relationship between this effect and social media (lines 251-255).

COMMENT: FRAMING OF THE QUESTION

I’d like to see a little more justification for the framing of the question and how it was presented to the participants. The authors mention in the discussion that “Of course, agreement with statements about animals making appropriate pets, or statements about being interested in pet ownership, does not equate one-to-one with actual pet ownership.” Is there any evidence from other papers or from this one of how this relationship works, or are there techniques to be able to account for the possibility of different interpretations? I think the question is problematic for me because although it seems really straightforward, it could be interpreted in different ways. For example: “I would like to have a python as a pet” seems pretty easy to answer on an operational basis, but it could also be interpreted on a more theoretical level. If some kind of negative control were included that were not operationally feasible in most instances,that could tell us something about the level of ‘realistic’ answers obtained.

For example: “I would like to have a rhinoceros as a pet” or “I would like to have a unicorn as a pet” might seem facetious as they’re operationally (or completely) impossible for most people but there might be a baseline level of people who would still see a picture of one and say “I want a rhino as a pet”.

I ask this and wonder if it’s possible to calibrate the results in this way because having spent time in zoos you do hear those kinds of comments, but I struggle to believe people really mean that they actually want a pet rhino/elephant/etc (although people constantly surprise me).

OUR RESPONSE: Yes, great point. Unfortunately we cannot identify any previous work that has quantified the relationship between interest and action, so we cannot figure out a way to calibrate our results as you’ve considered. However, we now expand upon the need for this data and discuss the value of considering the difference between stated and actual pet ownership (lines 272-284). We also reviewed our manuscript to ensure we are consistently referring to “interest in” and “perceptions of” rather than actual animal ownership to be clear we are talking about the psychology of interest in non-domestic pet ownership rather than the behavior of owning a non-domestic pet. 

COMMENT: LEVEL OF POSITIVE RESPONSES

Regardless of exactly how the question is interpreted, I think it is important to point out more clearly that the absolute interest in these animals as pets was low regardless of context (~20% for reticulate pythons in particular). This is important for understanding and managing demand – if it is a relatively small amount of the population that means measures to restrict sales/education potential buyers can be more focused than if the proportions were higher.

OUR RESPONSE: Interesting perspective! The coauthors felt that the interests reported here were high, rather than low. Even the lower python number of 20% is 1 of 5 people surveyed. Because there is no way to transform interest to action, going back to your comments above, we have opted not to interpret is as high or low, but rather to provide the numbers in the context of past research with other species (lines 228-234).

COMMENT: MORE DETAIL ON WILDLIFE TRADE

I think at present the way the wildlife trade for non-domesticated animals is too coarse. It’s a really complex, subtle subject matter, and the way the manuscript outlines on numerous occasions that its negative effects are clear/well-documented could contain more relevant detail to help the reader understand what those negative impacts are. Below I’ve listed some examples that I think would benefit from more detail. The quotation from the paper is in speech marks, my response/suggestion next to it.

“A considerable proportion of non-domesticated pets are sourced from wild populations, threatening biodiversity and conservation efforts [1, 2, 4-8]”.

Increasingly in many countries, the most commonly kept animals that are legally kept are from captive bred sources. What proportion is a considerable proportion (varies by region probably?)? Can you give any examples that are relevant to the current analyses?

OUR RESPONSE: We agree and have added specificity where possible (see below). We have rephrased this admittedly problematic sentence (line 59).

COMMENT: “Welfare, or the quality of life as experienced by the animal, is threatened when animals are captured from the wild, both during transport into captivity, and while they are in private homes or poorly regulated business ventures (e.g., animal cafes or touristic photo opportunities)”.

Key point for me: not only from wild-caught animals either – this is the case for any animal in the trade, including captive-bred ones. In fact, for many species in the wildlife trade in many countries the trade doesn’t necessarily cause conservation concern, but it still contains many welfare issues. For example, CITES listed species, although not perfectly monitored are generally sustainably managed, and species that aren’t CITES-listed in most cases are not listed because their population numbers aren’t threatened by trade. However, there are other issues here such as species becoming invasives and damaging native ecosystems, spread of pathogens, and in my mind most commonly and importantly, regardless of the conservation status of a species welfare issues at many points in the supply chain are a concern. I’d like to see more detail in the statements on “wildlife trade is bad” to provide evidence of how it’s bad, and in what contexts (ideally balanced with some examples of why it’s popular and also has benefits (the cost-benefit of any animal ownership, domesticated or not has many issues, not least welfare)).

[this reference has some good balanced arguments on the non-domestics trade: Pasmans, F., Bogaerts, S., Braeckman, J., Cunningham, A.A., Hellebuyck, T., Griffiths, R.A., Sparreboom, M., Schmidt, B.R. and Martel, A., 2017. Future of keeping pet reptiles and amphibians: towards integrating animal welfare, human health and environmental sustainability. Veterinary Record, 181(17), pp.450-450.]

OUR RESPONSE: We agree with most of this comment, and have tried to convey that non-domestics can experience welfare compromise regardless of origin, and have integrated reference to Pasmans et al. to call attention to an alternative route forward (line 60). We don’t see the opportunity to integrate discussion of the benefits of pet ownership in the current manuscript in a responsible way or dive deeper into the additional issues about wildlife trade, but hope that our revised wording in this section and others has done more justice to the complexity of the issues. 

COMMENT: “Furthermore, legal or illegal ownership of non-domesticated animals poses risks of disease transmission that are harmful both to humans and nonhuman animals”.

As above - this risk is almost certainly greater in the ownership and transport of domesticated animals. That doesn’t negate the above point, but needs acknowledging.

OUR RESPONSE: We agree and have revised to refer to the conditions of transport and housing rather than to certain categories of animals (line 64).

COMMENT: “their demand in the pet trade is negatively impacting wild populations [50, 51], and individuals of both species are likely to experience poor welfare in the private pet trade [51, 52].”

Reference 52 is about ball pythons, not reticulated pythons. These species are greatly different in their geographic range (West Africa Vs. S.E. Asia), and size (1.2-2m Vs 3m+). That being the case, the reference to poor welfare needs changing as these are different species with very different processes in the trade.

Further, there is actually empirical evidence that reticulated pythons can be managed for general trade in a sustainable fashion – see Natusch, D.J., Lyons, J.A., Riyanto, A. and Shine, R., 2016. Jungle giants: assessing sustainable harvesting in a difficult-to-survey species (Python reticulatus). PLoS One, 11(7), p.e0158397. for details

Again, that doesn’t mean there aren’t welfare issues with reticulated pythons in the trade or the hobby (they are very large animals that have specific husbandry needs), but the above points don’t sufficiently support the point being made.

OUR RESPONSE: Thank you, we have removed the reference concerning ball pythons and refer more generally to difficulties of supporting good welfare (line 125-127). Thank you for calling attention to Natusch et al 2016. This paper calls attention to assumptions about the impact on wild population harvests that we will keep in mind. We do not add citation this this since it is an analysis of harvesting for leather rather than pet trade, but do soften our assertion that the demand in the pet trade is negatively impacting wild populations (line 125).

Reviewer #3 

COMMENT: After reading the paper, I have recommended that it be resubmitted pending minor alterations. While the conclusions drawn about the variation between males and females is justifiable, it is my opinion that the discussion pertaining to the difference between generations requires expansion. As it is the primary focus of the paper, as stated in the paper title, the reasons for the results seen need to be presented in further detail. Considering the use of animals in media for decades, as well as other means of contact (circus' etc.), older generations would also have been exposed to substantial portrayals of non-domesticated animals in unnatural settings. It is for this reason that I recommend that additional explanations be presented to strengthen the point.

OUR RESPONSE: Thank you for your positive appraisal and this point. We agree, and in response to this comment and the similar comments of other reviewers, we have expanded upon the implications of the generational effect, see lines 248-269. We have also modified the title and moved the reporting of the generational effect earlier in the abstract and discussion to reflect this increased emphasis. We also added more information about the generational effect to Table 1.

COMMENT: Minor issues:

Figures 2 and 3 are blurred and subsequently, difficult to read.

When referring to Figures 1, 2 and 3 in the text, they ought to be referred to specifically (e.g. Fig. 3A) when discussing specific elements of the figure.

OUR RESPONSE: We have checked the resolution of the images and confirmed the resubmitted versions should be clear. We have updated references to the figures to refer to specific panels. 

COMMENT: Overall, I thought that the paper touched on a really interesting and important issue, and provides a strong foundation for the continuation of the investigation. I particularly appreciated the species chosen, to reflect the potential difference between animals with and without anthropomorphic characteristics.

OUR RESPONSE: Thank you.

---

## [Editor Report · Decision Letter 1]

20 Dec 2021

Younger generations are more interested than older generations in having non-domesticated animals as pets

PONE-D-21-28115R1

Dear Dr. Cronin,

We’re pleased to inform you that your manuscript has been judged scientifically suitable for publication and will be formally accepted for publication once it meets all outstanding technical requirements.

Kind regards,

Simon Clegg, PhD

Academic Editor

PLOS ONE

Additional Editor Comments

Many thanks for resubmitting your manuscript to PLOS One

As you have addressed all the comments and the manuscript reads well, I have recommended it for publication

You should hear from the Editorial Office shortly.

It was a pleasure working with you and I wish you the best of luck for your future research

Hope you are keeping safe and well in these difficult times

Thanks

Simon

---

## [Editor Report · Acceptance letter]

6 Jan 2022

PONE-D-21-28115R1 

Younger generations are more interested than older generations in having non-domesticated animals as pets 

Dear Dr. Cronin:

I'm pleased to inform you that your manuscript has been deemed suitable for publication in PLOS ONE. Congratulations! Your manuscript is now with our production department. 

Kind regards, 

on behalf of

Dr. Simon Clegg 

Academic Editor

PLOS ONE